
# 1 Analysis of the MODIS Above-Cloud Aerosol Retrieval

# 2 Algorithm Using MCARS

**Galina Wind[1,2], Arlindo M. da Silva[2], Kerry G. Meyer[2], Steven Platnick[2] and Peter**
**M. Norris[3,2]**
[1] SSAI, Inc. 10210 Greenbelt Road, Suite 600, Lanham, Maryland 20706, USA
[2] NASA Goddard Space Flight Center, 8800 Greenbelt Rd. Greenbelt, Maryland, 20771,
USA
[3] Universities Space Research Association, 7178 Columbia Gateway Dr., Columbia, MD
21046, USA
Correspondence to: G. Wind (Gala.Wind@nasa.gov)



**Abstract**
The Multi-sensor Cloud and Aerosol Retrieval Simulator (MCARS) presently produces

15synthetic radiance data from Goddard Earth Observing System version 5 (GEOS-5) model

16output as if the Moderate Resolution Imaging Spectroradiometer (MODIS) was viewing a

17combination of atmospheric column inclusive of clouds, aerosols and a variety of gases  and

18land/ocean surface at a specific location. In this paper we use MCARS to study the MODIS

19Above-Cloud AEROsol retrieval algorithm (MOD06ACAERO). MOD06ACAERO is

20presently a regional research algorithm able to retrieve aerosol optical thickness over clouds,

21in particular absorbing biomass burning aerosols overlying marine boundary layer clouds in

22the Southeastern Atlantic Ocean. The algorithm's ability to provide aerosol information in

23cloudy conditions makes it a valuable source of information for modeling and climate studies

24in an area where current clear sky-only operational MODIS aerosol retrievals effectively have

25a data gap between the months of June and October. We use MCARS for a verification and

26closure study of the MOD06ACAERO algorithm.

Our simulations indicate that the MOD06ACAERO algorithm performs well for marine

28boundary layer clouds in the SE Atlantic provided some specific screening rules are observed.

29For the present study, a combination of five simulated MODIS data granules was used for a

30dataset of 13.5 million samples with known input conditions. When pixel retrieval uncertainty

31was less than 30%, optical thickness of the underlying cloud layer was greater than 4 and

32scattering angle range within the cloud bow was excluded, MOD06ACAERO retrievals

33agreed with the underlying ground truth (GEOS-5 cloud and aerosol profiles used to generate

34the synthetic radiances) with a slope of 0.913, offset of 0.06, and RMSE=0.107. When only

35near-nadir pixels were considered (view zenith angle within +/-20 degrees) the agreement

36with source data further improved (0.977, 0.051 and 0.096 respectively). Algorithm closure

37was examined using a single case out of the five used for verification. For closure, the





38MOD06ACAERO code was modified to use GEOS-5 temperature and moisture profiles as

39ancillary. Agreement of MOD06ACAERO retrievals with source data for the closure study

40had a slope of 0.996 with offset -0.007 and RMSE of 0.097 at pixel uncertainty level of less

41than 40%, illustrating the benefits of high-quality ancillary atmospheric data for such

42retrievals.



## 1 Introduction

The MODerate resolution Imaging Spectroradiometer (MODIS) (Barnes et al., 1998) has proven to be an important sensor for aerosol data assimilation purposes for models such as the Goddard Earth Observing System Model, Version 5 (GEOS-5; Rienecker et al. 2008, Molod et al. 2012). There are two MODIS instruments on board NASA's Earth Observing System (EOS) *Terra* and *Aqua* spacecraft. There is a wide variety of data products available from these instruments for Land, Ocean and Atmosphere disciplines. Atmosphere discipline products include cloud mask, cloud top properties, cloud optical and microphysical properties and atmospheric aerosol properties. The MODIS data product files use a designation of MOD for Terra MODIS and MYD for Aqua MODIS. In this paper for brevity we will use "MOD" to refer to both instruments.

The largest contributor of biomass burning aerosols is Southern Africa (Reid et al, 2009, van der Werf et al, 2010). Biomass burning occurring from June through October creates thick smoke plumes that extend over the adjacent Atlantic Ocean. Prevailing winds in the area transport the smoke over the Southeast Atlantic Ocean (SEAO) and then as far as the Americas (Swap et al., 1996). The same time period coincides with a near-persistent layer of marine boundary-layer (MBL) stratus cloud that extends for several hundred miles westward from the Namibian coast (Devasthale and Thomas, 2011). The MODIS Dark Target aerosol retrieval algorithm (MOD04) that is used for ocean retrievals operates in clear sky conditions only. MOD04_DT retrievals are not provided for each individual MODIS pixel-level, but rather are performed over a 3x3 or 10x10 set of pixels. Moreover aerosol properties are not retrieved over sun glint regions (Kaufman et al, 1997, Levy et al, 2009, 2013). The SEAO region has both extensive seasonal cloud cover and a significant portion of MODIS granules



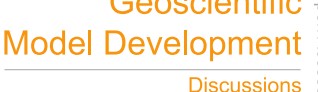

containing sun glint, leading to equally extensive loss of continuous observations from the
area.
Figure 1 illustrates these conditions using Terra MODIS data from 2006 through 2013.
Panel a) shows the percentage of ocean gridboxes in the SEAO area that had daily mean cloud
fraction greater than 50% in the MODIS Daily Level-3 gridded product (add reference) stored
at 1x1 degree resolution. Here, the SEAO area is defined the same way as in Meyer et al
(2015), specifically between -20 and +20 degrees longitude and +4 to -20 degrees longitude.
As much as 60% of all ocean gridboxes have cloud fraction greater than 50% in June (day
152) and only increase to the end of September (day 304). A 1-degree resolution gridbox will
contain some clear sky and thus at least some aerosol retrievals are possible. As shown in
Figure 1 b), in June about 70% of all ocean gridboxes contain some aerosol retrievals, though
by September that number drops to about 30-40%.
Due to aforementioned limitations of the standard dark-target MODIS aerosol algorithm,
a model that assimilates aerosol data from SEAO would have very few aerosol retrievals over
the ocean available to it. Most of the transport mechanism in the model would be thus
governed by the model physical processes (e.g., advection, sedimentation and wet removal
and vertical transport) instead of being constrained by observations.
The MOD06ACAERO algorithm (Meyer et al. 2015) fills in the aerosol data gap in
SEAO as it is able to perform retrievals of aerosol properties above MBL clouds. The
algorithm has been evaluated against observations from the Cloud-Aerosol Lidar and Infrared
Pathfinder Satellite Observation (CALIPSO) (Winker et al, 2009), but CALIPSO only
provides data at nadir and with a very limited spatial coverage. Recent improvements in
CALIPSO version 4 aerosol products (Kim et al, 2018) indicate that the comparisons shown
of the MOD06ACAERO algorithm with CALIPSO in Meyer et al (2015) would improve





91somewhat as significant work had been done to remedy the low bias that CALIPSO retrievals

92have. However, Kim et al (2018) state that the remaining SEA low bias in CALIPSO

93retrievals of AOD with respect to AERONET and MODIS makes CALIPSO retrievals

94somewhat problematic as means of aerosol algorithm evaluation for SEAO area. (e.g., Meyer

95et al, 2013, 2015, Jethva et al, 2014). Observations collected during the ObseRvations of

96Aerosols above CLouds and their IntEractionS (ORACLES) (Redemann et al, 2019) are

97currently being used to evaluate the MOD06ACAERO algorithm.

In this study we applied an Observing System Simulation Experiment (OSSE) framework

99to gain insight on the performance of the MOD06ACAERO algorithm. Rather than using the

100classic analysis/forecast error metric common in Numerical Weather Prediction OSSE studies

101(e.g., Hoffman and Atlas 2016) we adopt here a "Retrieval OSSE" perspective where the

102quality of the retrieval is used as the verification metric (Wind et al. 2013, 2016). A radiative

103transfer code is applied to the model quantities combined with sensor geometry to simulate

104how a model scene appears to a specific instrument. A retrieval algorithm designed for that

105instrument can be executed on the simulated measurements. Physical quantities retrieved by

106the algorithm can be compared to the known simulation input. The algorithm can be examined

107for closure over a large spatial domain and thus any areas or conditions that may be

108problematic for the algorithm could be examined, and the strengths and limitations of the

109algorithm can be extensively documented.

The Multi-sensor Cloud and Aerosol Retrieval Simulator (MCARS) is a tool that

111combines model output with a radiative transfer code in order to simulate radiances that may

112be measured by a remote sensing instrument if it were passing over the model fields (Wind et

113al, 2013, 2016). In this paper, MCARS continues to use the combination of the GEOS-5

114model, correlated-$k$ models of atmospheric transmittance due to various gaseous absorbers for



MODIS channels as per Kratz (1995), inline Rayleigh scattering and the Discrete Ordinate
Radiative Transfer (DISORT) code (Stamnes et al. 1988) to simulate MODIS radiances. Two
improvements have been made to the MCARS code since last publication. The computational
resolution has been increased to 32 streams, up from 16. Additionally, for this study the
higher resolution 7 km GEOS-5 Nature Run (G5NR) was used in place of the standard 25 km
resolution GEOS-5 output (Gelaro et al. 2015, da Silva et al. 2015, Putman et al. 2015).
G5NR is a 2-year global, non-hydrostatic mesoscale model dataset for the period 2005-2006
produced with the GEOS-5 Atmospheric GCM. The model run is performed at a horizontal
resolution of 7 km using a cubed-sphere horizontal grid with 72 vertical levels, extending up
to 0.01 hPa (~ 80 km). In addition to standard meteorological parameters (wind, temperature,
moisture, surface pressure), this GCM includes 15 aerosol tracers (dust, sea-salt, sulfate, black
and organic carbon), $O_3$ and $CO_2$. The GEOS-5 NR is driven by prescribed sea-surface
temperature and sea-ice, daily volcanic and biomass burning emissions, as well as high-
resolution inventories of anthropogenic sources. A description of the GEOS-5 model
configuration used for the Nature Run can be found in Putman et al. (2014), while results
from a validation exercise appear in Gelaro et al. (2015) and Castellanos et al. (2019).

In a previous study of the MOD04_DT code (Wind et al, 2016), we had the advantage of

having simultaneous in situ aerosol property measurements from AErosol RObotic NETwork
(AERONET) (Holben et al., 1998). AERONET has very limited data available over ocean,
mainly from islands and ship transits. Even in places where AERONET is established, no
measurements can be obtained in presence of clouds. Therefore, no ground-based in-situ
measurements can be included in our analysis of the MOD06ACAERO product and so the
analysis is necessarily limited to verification and closure.



In sections that follow we will describe the application of MCARS to study the

139MOD06ACAERO algorithm. Section 2 very briefly describes the MCARS code and the

140experiment setup. Section 3 describes the MODIS MOD06ACAERO product of Meyer et al.

141(2015). Section 4 shows the details of the study and study conclusions. Finally, section 5

142discusses the next steps in MCARS development.

## 1432   MCARS description

The MCARS code was previously described in detail in Wind et al (2013, 2016).

145Therefore, only a brief description will be given here. Global aerosol, cloud, surface and

146atmospheric column fields from the G5NR simulation as described above serve as the starting

147point for radiance simulations. The GOCART bulk aerosol scheme currently used in the

148G5NR is used for the simulations reported in this paper, with corresponding optical properties

149as described in Randles et al. (2017), Hess et al (1998)  and references within. The simulation

150input data was produced in accordance with the methods outlined in Wind et al. (2016). The

151G5NR model output was split into 1-km subcolumns (MODIS pixel resolution) using the

152independent column approximation method as described in detail in Wind et al. (2013). Here

153a brief summary of the model data preparation methodology is given.

MODIS pixels for each GEOS-5 gridbox were collected and the same number of pixel-

155like sub-columns was generated using a statistical model of sub-gridcolumn moisture

156variability. The sub-column generation used a parameterized probability density function

157(PDF) of total water content for each model layer and a Gaussian copula to correlate these

158PDFs in the vertical (Norris et al, 2008, Norris and da Silva 2016a,b).

The subcolumns generated in this way were subsequently rearranged, to give horizontal

160spatial coherence, by using a horizontal Gaussian copula applied to condensed water path.

161This arrangement had to be applied in order to create spatially coherent cloud-like structures.



162The subcolumns themselves were not altered in any way during this process. If this step is

163skipped and the subcolumns are placed randomly within each gridbox the MODIS Cloud

164Optical and Microphysical Properties (MOD06) product (Platnick et al, 2017) would restore

165many of the pixels to clear sky unless the initial gridbox had close to 100% cloud fraction

166(Zhang and Platnick 2011; Pincus et al. 2012). The MOD06 product is a necessary input for

167MOD06ACAERO and must be produced prior to MOD06ACAERO execution. The need for

168this subcolumn rearrangement is significantly lessened when G5NR is used because the

169smaller gridboxes are often close to 100% cloudy especially in MBL regimes, but removing

170the method from the model preparation step was not practical due to its small impact on

171execution time and possibility of introducing errors.

The layer aerosol properties were obtained using the independent column approximation

173with the same PDF of total water content as used for clouds. A GEOS-5 aerosol species

174output file was used in conjunction with aerosol optical properties as in Randles et al. (2017).

175The aerosol phase functions for each of the 15 species output by GEOS-5 were produced and

176combined on the fly to create a single bulk set of scattering properties and Legendre

177coefficients. (Wind et al, 2016)

Model parameters such as profiles of temperature, pressure, ozone and water vapor

179together with layer information about clouds and aerosols are combined with solar and view

180geometry of the MODIS instrument. Surface information is also a combination of GEOS-5

181information of surface temperature, snow and sea ice cover and MODIS-derived spectral

182surface albedo (Moody et al. 2007, 2008). All of these parameters are transferred to the

183DISORT-5 radiative transfer code and reflectances and radiances in 22 MODIS channels

184between 470nm and 14.2μm are produced. The default computational resolution of DISORT-

1855 has also been increased to 32 streams up from 16 used in the two previous studies.

186Additionally some of the simulations in this study were executed at 64 streams. Final MCARS



output is packaged in a format identical to the standard MODIS Level-1B radiometric files
and is thus completely transparent to any operational or research-level retrieval algorithm
code.
These simulations were produced at the NASA Center for Climate Simulations (NCCS)
supercomputer. Each complete simulation of a MODIS-like granule requires 5.5 hours of wall
clock time on 300 processors. Computational throughput can be increased by limiting the
scope of the simulation to fit a particular investigation. For this study, however, we retain the
full set of channels needed for both cloud and aerosol research.

**3   MODIS above-cloud aerosol properties product**

The MODIS above-cloud aerosol properties product (MOD06ACAERO) (Meyer et al.
2015) is a regional algorithm able to simultaneously retrieve MBL cloud optical thickness
(COT), cloud effective radius, and aerosol optical depth (AOD) above-cloud in the SEAO
region. It uses six MODIS channels (bands 1-5 and 7) having central wavelengths of 0.47,
0.55, 0.66, 0.86, 1.24 and 2.1μm. The MOD06ACAERO algorithm takes advantage of the
strong biomass burning aerosol absorption gradient in the visible (VIS) to near-infrared (NIR)
spectrum that, when the aerosol layer overlies a bright cloud, yields differential attenuation
(stronger at shorter wavelengths) of the otherwise nearly spectrally invarient top-of-
atmosphere cloud reflectance across the VIS/NIR. Sensitivity to cloud optical thickness is
localized in the spectral range between 0.47 and 1.24μm and is directly related to the
magnitude of reflectance, while sensitivity to above-cloud aerosol optical depth is related to
the spectral slope of the reflectance. The MOD06ACAERO algorithm uses 2.1μm channel for
cloud effective radius information. That is also consistent with the principal retrieval
contained in the MOD06 product (Platnick, et al, 2017)

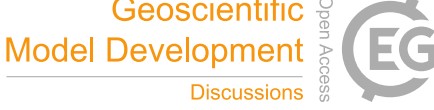

The MOD06ACAERO retrieval inversion uses an optimal estimation-like approach

213(Rodgers, 1976) that attempts to minimize the difference (cost function) between the six

214MODIS reflectance observations and forward-modeled reflectance that is a function of cloud

215optical thickness, effective radius, and above-cloud AOD. However, rather than in-line

216radiative transfer calculations, MOD06ACAERO relies on a set of pre-computed lookup

217tables (LUTs) of coupled cloud and above-cloud aerosol reflectance. These LUTs are

218generated using the same cloud microphysics models used by MOD06 (Platnick et al, 2017)

219and the absorbing aerosol model used by MOD04_DT over land surfaces (Levy et al, 2013).

220Retrievals using a second aerosol property model, one based on field campaign data from

221SAFARI 2000 (Haywood et al, 2003), are also available in MOD06ACAERO output. While

222these Haywood et al. model retrievals were recommended in Meyer et al (2015), evaluation

223during the ORACLES campaign revealed deficiencies at certain scattering angle ranges (K.

224Meyer, private communication). Thus, for this study we use the MOD06ACAERO results

225based on the MOD04_DT aerosol models.

The MOD06ACAERO retrieval operates at 1km resolution, compared to the 10km and

2273km MOD04_DT resolutions, and simultaneously provides pixel-level estimates of retrieval

228uncertainty accounting for known and quantifiable error sources (e.g., radiometry,

229atmospheric profile errors, cloud and aerosol forward model errors) consistent with the

230MOD06 cloud product methodology (Platnick et al, 2020). Figure 2 shows an example

231retrieval result from MOD06ACAERO compared to MOD04_DT standard 10km output. The

232Terra MODIS granule shown here, from 2006 day 224 at 10:05 UTC, has extensive cloud

233cover over the ocean, typical for this season. MOD04_DT provides a very limited amount of

234data, localized to the few areas of clear sky, while MOD06ACAERO fills in the above-cloud

235area.

MOD06ACAERO uses National Center for Environmental Prediction (NCEP)



atmospheric profile products (Derber et al, 1991) for atmospheric correction. As part of our
investigation we will look at impact of discrepancies between NCEP and G5NR on retrieved
aerosol properties.

## 241 **4    Analysis**


To create the data used for the MOD06ACAERO verification study, we examined the
G5NR dataset for cases that were similar to conditions commonly encountered during the
burning season over SEAO. August 2006 was selected because it was a very active smoke
season and a significant amount of MBL clouds were present in the model output. Models
often have difficulties forming MBL clouds as higher than usual grid and vertical resolution is
needed in order to accurately represent the processes that lead to MBL formation in nature.
As real Terra and Aqua overpasses are needed in order to define the sun-satellite
geometry for the MCARS simulations, satellite orbital tracks had to be considered. Because
orbital gaps are prominent in the MODIS data over the SEAO MBL region, care must be
taken in selecting specific days and times having adequate sensor geometry. Technically
because MCARS is a simulation, orbital gaps have no meaning. But because of the need of
actual sensor geometry to start the simulation, it is most expedient to simply browse available
MODIS data for a suitable track. Even though G5NR does not perform any data assimilation,
the model code is identical to the standard GEOS-5 model. MCARS normally runs on
standard GEOS-5 output. In Wind et al (2013) we showed MCARS as a model output
verification tool. It is always very desirable to match date/time/orbit when model performance
may be compared to real concurrent sensor measurements. Even though no orbital match is
required in this study, a decision was made to not alter the standard MCARS operation in
order to avoid accidental introduction of software issues. Five cases were selected under these





262considerations. Three came from Terra MODIS overpasses and two from Aqua MODIS. The

263times and dates were as follows. Terra MODIS: 2006 day 224, 10:05 UTC, 2006 day 225

26409:10 UTC, 2006 day 228 09:40 UTC. Aqua MODIS: 2006 day 224 12:55 UTC and 2006 day

265226 12:40 UTC. This dataset comprises 13.5 million points where the atmospheric column

266and surface conditions are explicitly known.

Figure 3 a) shows simulated RGB images for the 5 MCARS MODIS granules listed

268above. Also shown in b) are the same simulated granules where the aerosols have been

269removed from the radiative transfer simulations. This ability to remove clouds, aerosols or

270gases from from the simulation offers extensive control evaluating the performance of

271retrieval algorithms and diagnosing algorithm deficiencies.

There is a significant similarity between the real Terra MODIS granule of Figure 2 and

273the simulated granule for the same date and time. The G5NR is a free running model and does

274not perform any data assimilation, and therefore it is not synoptically locked to the particular

275day depicted in Figure 2. The apparent similarities between Figures 2 and 3 merely reflect the

276persistent patterns of MBL clouds and smoke in the region. There is no expectation of a match

277with any real data in this study. It is not a statement to G5NR performance as in other cases

278the cloud amount/distribution had no match to any real data. It is merely an interesting

279coincidence. Some granules were selected to include a significant portion of land surface for a

280later examination of the MOD04_DT retrievals, repeating the study in Wind et al (2016) in a

281different region (not reported here).

This dataset, both the complete and the clean (aerosol-free) versions, was fed through the

283standard operational MODIS Data Collection 6 cloud product processing chain to produce

284cloud mask, MOD06 cloud top and optical properties, and finally the MOD06ACAERO

285output for each case. Results from all granules were then combined and only retrievals for

286cloudy pixels were examined. The MOD06ACAERO aerosol retrievals were compared to



287source aerosol optical depth provided by GEOS-5 (Wind et al, 2013). Figure 4 shows results

288of this comparison. The only constraint on this comparison was that the algorithm-reported

289pixel-level retrieval uncertainty had to be less than 40% for panel a) and less than 30% for

290panel b). One of the motivations of this study was to characterize errors in the

291MOD06ACAERO algorithm for subsequent aerosol data assimilation into GEOS-5. Pixels

292with higher uncertainties could be considered in the analysis, but assimilating data where the

293retrieval error is 50% or greater could negatively impact the assimilated fields. As depicted in

294Figure 4, filtering retrievals at the reported algorithm uncertainty at 40% is very effective to

295produce a good match between MOD06ACAERO and the G5NR output variables, with the

296exception of very low AODs. G5NR uses aerosol models described in detail in Randles et al

297(2017). It is a set of 15 absorbers, properties of which are a function of column relative

298humidity. MOD06ACAERO in this study uses the MOD04_DT aerosol models, which are

299distinct in composition and additionally computed at a constant 80% column relative humidity

300(Levy et al, 2013). Because G5NR mixes aerosols on-the-fly to create bulk layer properties

301and MOD06ACAERO has a constant regional mixture, there is a natural source of uncertainty

302in any comparison of MOD06ACAERO retrievals with G5NR. However the regional mixture

303of MOD04_DT had been used extensively to train the GOCART model used by both GEOS-5

304and G5NR. Thus we expect the uncertainty due to aerosol model mismatch to be fairly

305minimal. Same exact situation of aerosol mixture mismatch exists in real data and is most

306likely greater than the one existing in this simulation.

Meyer et al. (2015) suggest that additionally MOD06ACAERO retrievals should be

308screened by retrieved cloud optical thickness and that they should be discarded if COT is less

309than 4.0. We applied this additional constraint onto the retrieval comparison and the result is

310shown in Figure 5. Discarding the AOD retrievals when cloud is thin improved the match-up

311against GEOS-5, but there still appears to be an issue when GEOS-5 AOD is very close to



312zero.

313 The power of MCARS lies in being able to tightly control simulation parameters. The 314MOD06ACAERO algorithm appears to run into a difficulty at low source AOD. In order to 315examine the causes for this discrepancy in more detail, we turn our attention to the clean 316MCARS case shown in figure 3b) by setting the AOD precisely to zero and examining the 317retrieval performance in such situation. Ideally MOD06ACAERO should retrieve a zero AOD 318throughout. With an exception of a narrow range of scattering angles between 135 and 145 319degrees, which corresponds to the cloud bow direction, the algorithm indeed retrieved AOD 320that was extremely close to zero. Figure 6 depicts the difference between retrieval and source 321as a function of scattering angle. Retrievals where MOD06ACAERO matched GEOS-5 322precisely were discarded for clarity. Within the cloud bow MOD06ACAERO tends to return a 323small positive AOD of about 0.15.

324 The liquid water phase function is very complex in the cloudbow region and is very 325difficult to model accurately. That particular region has consistently caused difficulties to the 326standard MOD06 product retrievals of MBL clouds. Both MOD06 and MOD06ACAERO 327LUTs are computed at 64 DISORT streams. We performed some investigation of this area by 328running a special simulation for a single case from Terra 2006 day 224 10:05 UTC. This case 329was selected because the cloudbow is especially noticeable in both real and simulated data. 330The simulation was also executed using 64 DISORT streams in order to reduce uncertainties 331associated with the simulation being performed at half the resolution. In cloudbow region 332more streams would potentially lead to a better model. Unfortunately the cloudbow persisted. 333It thus may be the case that 64 streams are not sufficient to properly resolve the cloudbow in 334either simulation or retrieval. Even higher resolution may be advisable. Increasing 335computational resolution of MOD06 LUTs is presently considered for the upcoming MODIS 336Data Collection 7. Depending on the results, same increase may occur for MOD06ACAERO.



337At this time, for purpose of establishment of assimilation constraints, which is the focus of
338this study, one might simply exclude the cloud bow scattering angle range from consideration
339until more is known.

Figure 7 shows the results of MOD06ACAERO retrievals from Figure 5 where  retrievals
341within the cloud bow have been discarded. The comparison with source data is further
342improved and the cluster of MOD06ACAERO retrievals present in Figure 5 when GEOS-5
343AOD was near zero has disappeared.

Often better retrievals can be obtained when less oblique view geometry is considered in
345real data. Pixel size, longer optical path length and 3D effects from clouds can all make
346retrievals performed at oblique view angles less optimal. In the case of this study, another
347consideration for imposition of a view zenith limit is that presently MCARS does not account
348for pixel size growth at oblique view angles. The number of subcolumns generated does not
349change with view zenith angle. Therefore, MCARS results when view angle is oblique may
350not be an accurate measure of algorithm performance as only the effects of optical path length
351are simulated.

The MOD06 cloud product outputs cloud top pressure, temperature and height limited to
353near nadir in addition to full swath products. The "near nadir" is defined as viewing zenith
354angle less than 32 degrees (Menzel et al, 2008). Figure 8 shows the MOD06ACAERO
355retrievals of Figure 7 further limited by view zenith angle of less than 32 degrees. When view
356zenith angle is limited to 32 degrees the comparison with GEOS-5 source data is again
357improved. We can now show a slope of 0.866 for retrievals with less than 40% error and
3580.913 for retrievals with error of less than 30%. Note that even though the data extent had
359been limited, there are still over 600,000 data points left to be ingested into a model if data
360assimilation were to be attempted in an area where previously the number of such data points
361was close to 0.



We can constrain the view zenith angle range even further as shown in Figure 9, reducing
363the threshold to 20 degrees. Whereas the comparison shows all around improvement with
364slope of 0.931 and 0.977 for retrieval error of less than 40% and 30% respectively, the number
365of points suitable for assimilation shrinks by half. It is not clear if this dataset size reduction
366can be justified by the improvement in alignment with the source data.

With the 20 degree view angle constraint the algorithm results are very close to source
368data and we could potentially state that we have closure against source GEOS-5 data even
369though both MOD06 and MOD06ACAERO run under operational conditions used NCEP
370GDAS data for atmospheric correction (implying a likely overestimation of the error in these
371profiles). In order to assess the impact of using these GDAS-based profiles we consider a final
372experiment where we use MCARS pixel-level input profiles for atmospheric correction. The
373result is shown in Figure 10. When atmospheric profiles are removed as a source of
374inconsistency, the agreement with source data improves to a slope of 0.996 with intercept of -
3750.007 and RMSE of 0.097 for retrievals with less than 40% error and slope of 0.989, intercept
376of 0.03 and RMSE of 0.085 for retrievals with less than 30% error. Small sample size for
377retrievals with lower uncertainty is the reason for somewhat lesser agreement with source data
378for this closure experiment. The remaining source of potential disagreement of
379MOD06ACAERO retrieval with input GEOS-5 data is the difference between aerosol models
380used by MCARS and MOD06ACAERO. Cloud models between MOD06ACAERO and
381MCARS are identical in this study. The MOD06ACAERO model is fixed for the region,
382while the GEOS-5 aerosols are fully dynamic as per Randles et al (2017). However, it is not
383practical to change either MCARS or MOD06ACAERO code to use a different aerosol model
384set, and with the agreement being as good as it presently is.






## 386 **5    Conclusions and future directions**

This paper is a direct evolution of work started in Wind et al, (2013) and continued in
Wind et al (2016). The Multi-sensor Cloud and Aerosol Retrieval Simulator (MCARS) has
now been applied as a verification tool for a research-level algorithm. The algorithm studied
was the MODIS above-cloud aerosol properties retrieval algorithm of Meyer et al (2015).
MCARS computational resolution has been doubled and for this study the high-resolution
(7km) GEOS-5 Nature Run model was utilized. The MCARS code produces radiances and
reflectances in a standard MODIS Level 1B format after sending the GEOS-5 data through
DISORT-5 radiative transfer code. The output can be directly ingested by any retrieval or
analysis code that reads data from the MODIS instrument.
We used the MCARS code to perform verification and closure study on the
MOD06ACAERO algorithm. In this study we generated a set of five MODIS granules located
in the Southeastern Atlantic Ocean off the coast of Namibia. We executed the
MOD06ACAERO code on this case set. In the verification part of the study the algorithm
performed very well. When pixels with less than 30% uncertainty were considered with
underlying cloud layer having optical thickness greater than 4 the algorithm matched the
source GEOS-5 aerosol optical depth with slope of 0.774 and offset of 0.076, RMSE = 0.131.
On further examination, executing the algorithm on the same case set with aerosols removed it
was determined that there might be data that is less useful around the scattering angle of 140
degrees, the cloud bow direction. When the cloud bow pixels were excluded the slope
improved to 0.913. The near-nadir slope with angle limit of 20 degrees improved the
agreement further to 0.977, RMSE=0.096.
To look at closure one of the five cases was selected. For closure both MOD06 and
MOD06ACAERO codes were modified to use MCARS input profiles as ancillary instead of
the NCEP analysis used in operations (Platnick et al, 2017). When the results were compared



to source GEOS-5 data a slope of 0.996 with offset of -0.007 and RMSE = 0.097 was reached
for pixels with less than 40% uncertainty. The agreement was slightly worse for uncertainties
less than 30% (slope 0.989, offset 0.03 and RMSE = 0.085) but that was mainly due to having
a smaller number of pixels in the set, only 130,000.
The results of this study suggest that retrievals produced by MOD06ACAERO are of
good initial quality and would be a valuable addition to model data assimilation streams with
the following constraints. MOD06ACAERO pixels should be assimilated if retrieval
uncertainly is less than 40%, if optical thickness of the underlying cloud layer is greater than
4.0 and if the pixel scattering angle is outside the cloud bow. Additionally, an even tighter
constraint can be added to only take pixels that are near nadir.
This study is yet another example of the capabilities of the MCARS framework. There
are many other potential applications of the MCARS code, including extending the simulator
to other sensors and examining the performance of fast retrieval simulators used in climate
modeling.

## 425 6   Code and Data Availability

The MCARS code and any datasets produced, including all data shown (GEOS-5 input
in netCDF4 and all MODIS output in HDF4 file format) and discussed in this paper, are
available to users free of charge by contacting the authors. There may be additional, wider
distribution means in the future as needed. We have not deemed it practical up to this time to
release the MCARS source code into general-purpose source repositories. The data files are
quite large with source input data being on the order of 20 Gb for each MODIS-like granule
created. The GEOS-5 model source code is publicly available, and we may release the
MCARS code under the same NASA Open Source Agreement and the same repository.



## Author Contributions

GW is the development and experiment design lead on the MCARS project. She maintains the code, creates experiments and performs most of the analysis of experiment data.

AdS and PN assist with preparation, interpretation and integration of the GEOS-5 model data.

KM is the author of MODIS above-cloud aerosol retrieval algorithm, the subject of this simulation experiment. He assisted with interpretation of retrieval results and development of assimilation constraints for the above-cloud aerosol product.

SP assisted with analysis, evaluation and interpretation of all experiment data.

## Acknowledgements

The authors would like to thank Brad Wind for the initial idea for creating a simulator, the output of which could be transparently used with remote sensing retrieval codes.



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



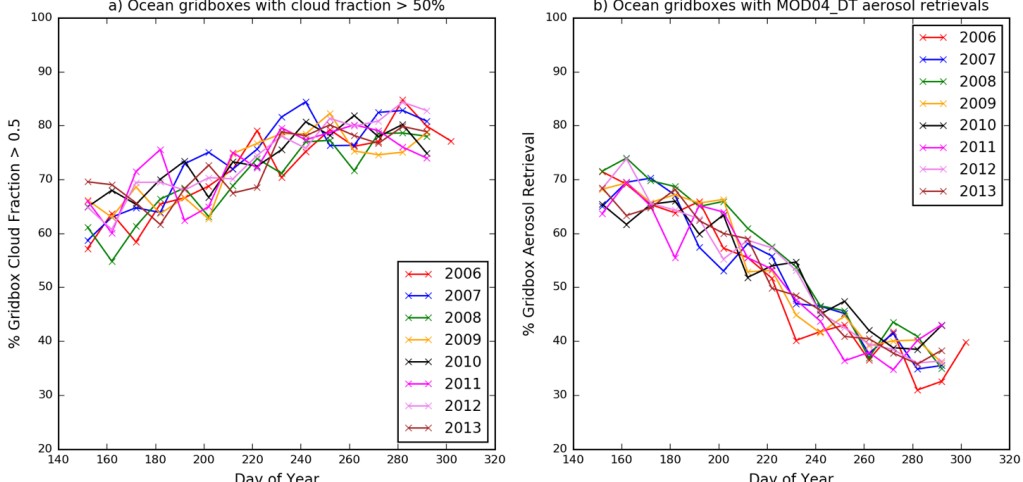

597Figure 1. Terra MODIS Level-3 Daily 1-degree gridded product for SEAO area for years
5982006-2013. Panel a) shows the percentage of SEAO ocean gridboxes that had cloud fraction
599greater than 50%. Panel b) shows the percentage of SEAO ocean gridboxes that had any
600successful MOD04DT aerosol property retrievals of any quality.






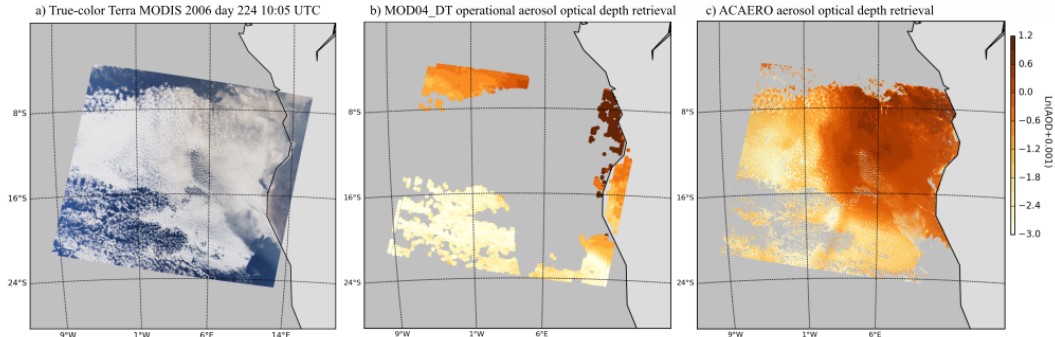


Figure 2. Real-data example of MOD06ACAERO retrieval. Terra MODIS 2006 day 224
10:05 UTC. Panel a) shows the true-color MODIS granule. There is extensive aerosol layer
above the equally extensive MBL cloud layer. Panel b) shows the MODIS Data Collection 6
operational Dark Target aerosol retrieval. It is a 10km resolution product with retrievals
available only in clear sky conditions and outside glint. Panel c) shows the MOD06ACAERO
above-cloud aerosol retrieval that is able to fill the data gap created by presence of MBL
layer.



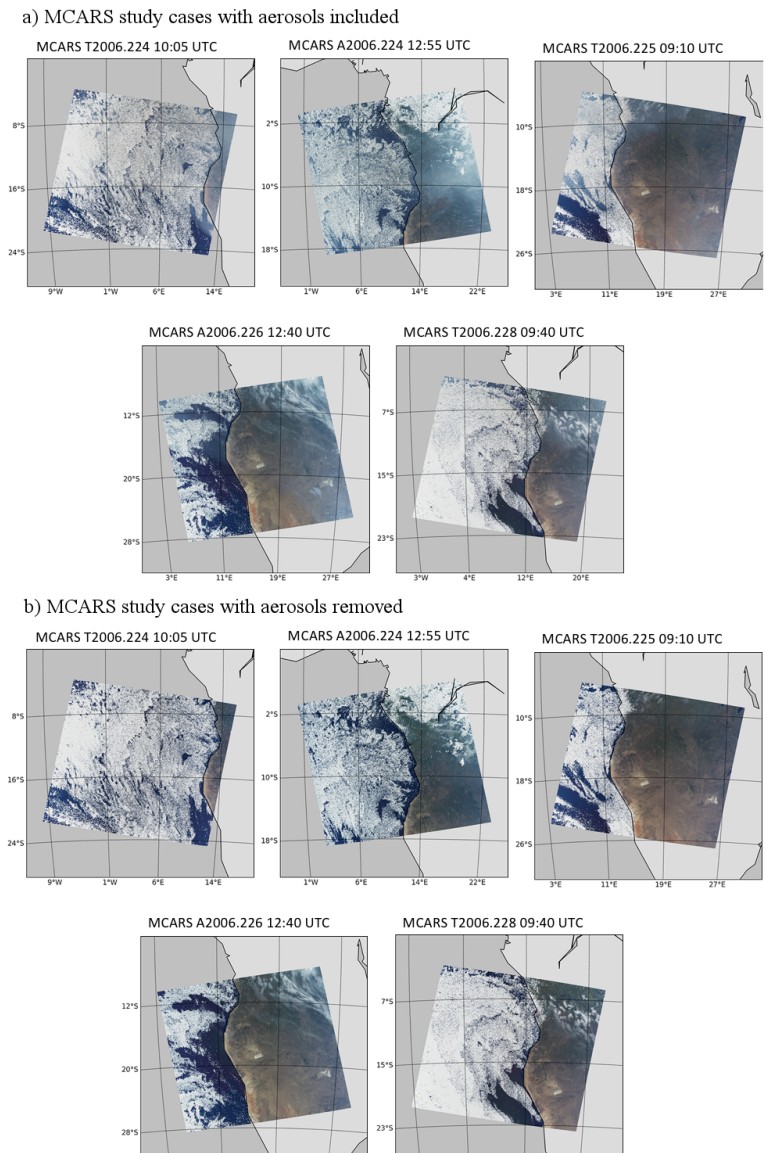


Figure 3. Scenes generated by MCARS from G5NR used in analysis of the MOD06ACAERO
product. There are three cases based on Terra MODIS, designated with a T next to the year.
There are two cases based on Aqua MODIS, designated with an A next to the year. Panel a)
shows the case set simulated with aerosols present. Panel b) shows the same case set but
simulated with aerosols removed.





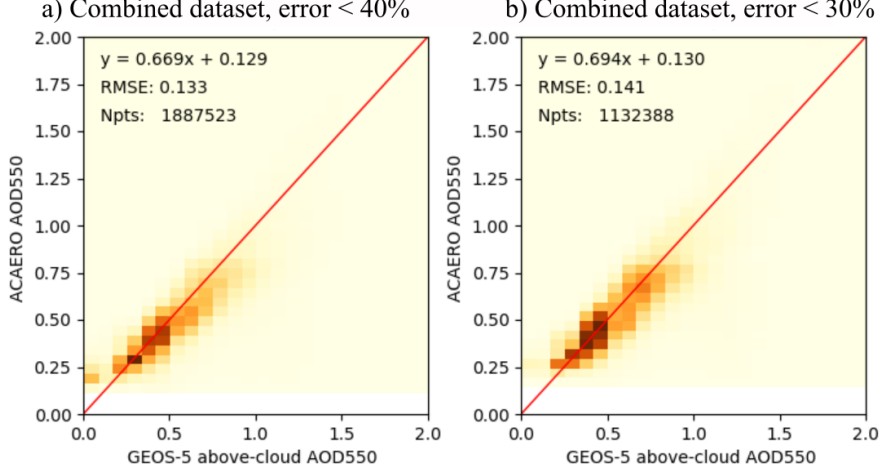


Figure 4. MOD06ACAERO retrieval results from the combined dataset of Figure 3a
compared to source GEOS-5 aerosol optical depth. No screening of retrievals had been
performed except for pixel-level uncertainty. Panel a) shows MOD06ACAERO retrievals
with uncertainty of less than 40% and panel b) shows same with uncertainty less than 30%.






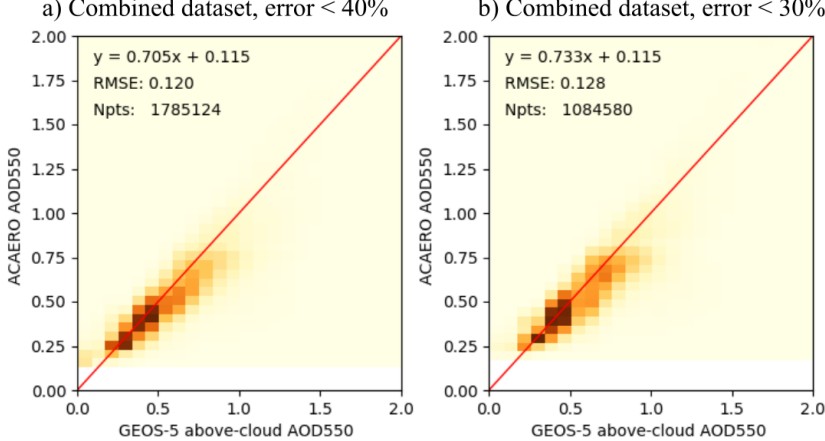

629Figure 5. MOD06ACAERO retrieval results from the combined dataset of Figure 3a
630compared to source GEOS-5 aerosol optical depth. AOD retrievals where COT was less than
6314 are now discarded. Panel a) shows MOD06ACAERO retrievals with uncertainty of less than
63240% and panel b) shows same with uncertainty less than 30%.



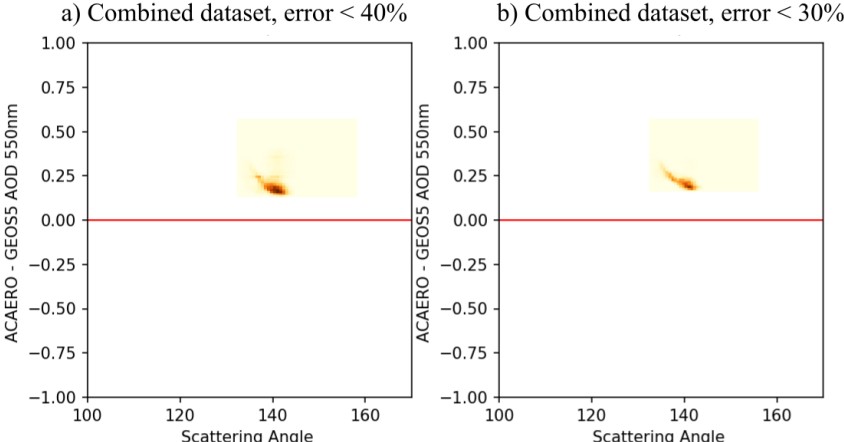

Figure 6. MOD06ACAERO retrieval results from the combined dataset of Figure 3b, where
aerosols had been removed. The results are displayed as difference from GEOS-5 AOD,
which in this case was zero, as a function of scattering angle. All retrievals where
MOD06ACAERO result was also zero had been removed for clarity. All non-zero
MOD06ACAERO retrievals appear to be concentrated in a narrow angle range between 135
and 145 degrees which corresponds to the cloud bow. Panel a) shows MOD06ACAERO
retrievals with uncertainty of less than 40% and panel b) shows same with uncertainty less
than 30%.



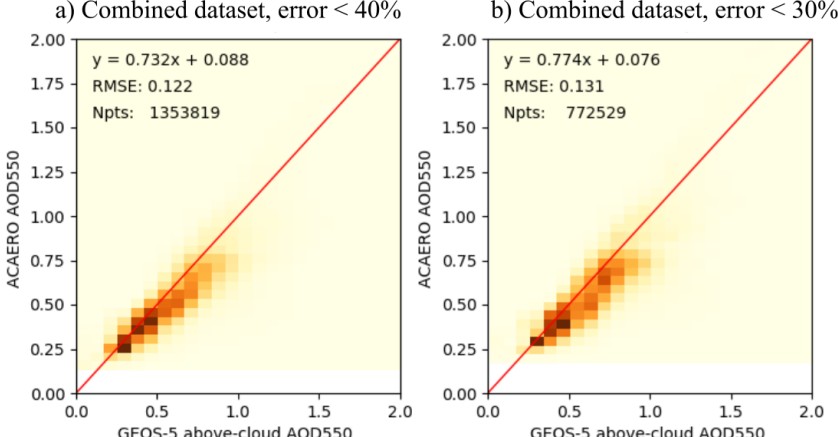

644Figure 7. MOD06ACAERO retrieval results from the combined dataset of Figure 3a
645compared to source GEOS-5 aerosol optical depth. AOD retrievals where COT was less than
6464 are now discarded. Additionally retrievals in the cloud bow region are also removed. It
647appears they were indeed the source of a cluster of higher MOD06ACAERO retrievals when
648GEOS-5 AOD was near zero and the match up with GEOS-5 source AOD is further
649improved. Panel a) shows MOD06ACAERO retrievals with uncertainty of less than 40% and
650panel b) shows same with uncertainty less than 30%.
651
652

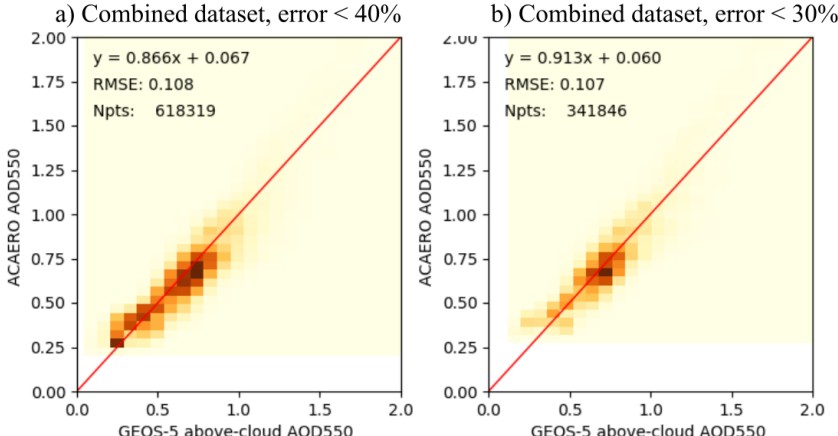

653Figure 8. MOD06ACAERO retrieval results from the combined dataset of Figure 3a
654compared to source GEOS-5 aerosol optical depth. AOD retrievals where COT was less than
6554 and where the scattering angle was in the cloud bow are now discarded. Additionally the
656data extent had been limited to only include pixels with view zenith angle of less than 32
657degrees. Retrieval comparison shows further improvement. Panel a) shows MOD06ACAERO
658retrievals with uncertainty of less than 40% and panel b) shows same with uncertainty less
659than 30%.

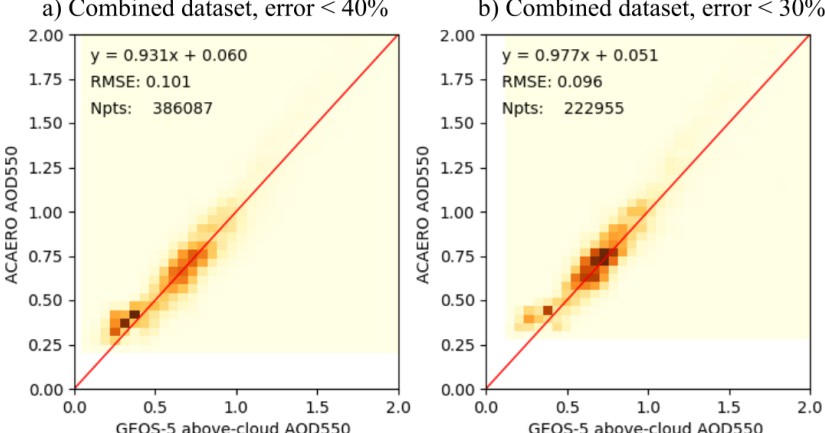

Figure 9. MOD06ACAERO retrieval results from the combined dataset of Figure 3a
compared to source GEOS-5 aerosol optical depth. AOD retrievals where COT was less than
4 and where the scattering angle was in the cloud bow are now discarded. Additionally the
data extent had been limited to only include pixels with view zenith angle of less than 20
degrees. Retrieval comparison shows further improvement however it is not clear if the
reduction in dataset size is worth the gain in accuracy. Panel a) shows MOD06ACAERO
retrievals with uncertainty of less than 40% and panel b) shows same with uncertainty less
than 30%.

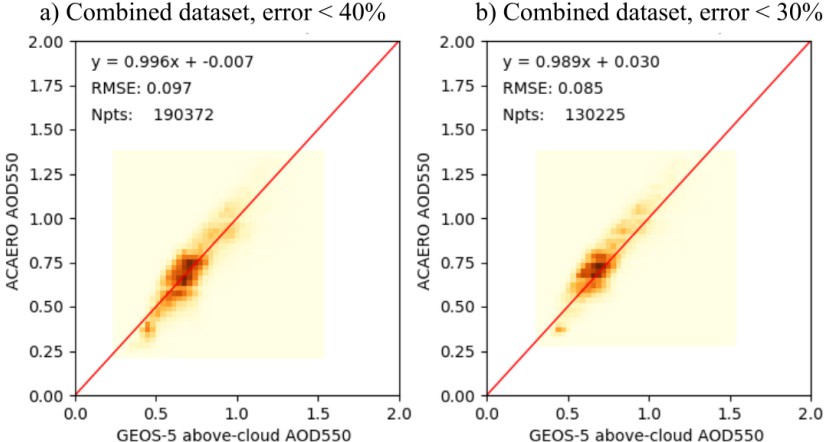

670Figure 10. MOD06ACAERO retrieval results from simulated MCARS granule based on Terra
671MODIS 2006 day 224 10:05 UTC compared to source GEOS-5 aerosol optical depth. In this
672experiment both MOD06 and MOD06ACAERO were modified to use MCARS pixel-level
673atmospheric profiles to perform atmospheric correction. AOD retrievals where COT was less
674than 4 and where the scattering angle was in the cloud bow are now discarded. Additionally
675the data extent had been limited to only include pixels with view zenith angle of less than 20
676degrees. This experiment shows excellent agreement with source data. Panel a) shows
677MOD06ACAERO retrievals with uncertainty of less than 40% and panel b) shows same with
678uncertainty less than 30%. The small dataset size in panel b) is the reason for slightly lower
679agreement with source compared to panel a)
680