# Peer review of "Analysis of the MODIS Above-Cloud Aerosol Retrieval"

_Geoscientific Model Development, 2021_

## Author Comment (AC2)

Summary:

This paper presents a method to quantify the accuracy of the MOD06ACAERO retrieval through the combination of a forward radiative transfer model built upon the GEOS-5 model. This manuscript is aimed at evaluating the MOD0ACAERO product for inclusion in a data assimilation scheme for GEOS-5. This aerosol assimilation would be quite useful considering the lack of continuous observations of aerosol over clouds in the South East Atlantic Ocean. The use of MCARS as a closure study illustrates quite well the needed filtering processes of MOD06ACAERO in order to be used as a data source for assimilation and puts both the MCARS and MOD06ACAERO in high regards. The data and methods are clearly defined, while the codes are not currently available in a public access repository, it is mentioned that it can be accessed by contacting the author.

The manuscript is very well written and is quite pertinent to model development, particularly GEOS-5. After review of the manuscript there are only very few minor revisions to be made, however one concern, which could be elevated to a major revision, regarding the underlying aerosol model and how it is represented in both the retrieval and GEOS-5 should be addressed before publication (see below). It is suggested that this manuscript is to be published with this one major revision.

Major Comment:

The MCARS application of building a retrieval OSSE on top of GEOS-5 seems to be very powerful, but there is limited discussion of the potential pitfalls of such an approach. Of immediate concern for interpreting the retrieval accuracy of MOD06ACAERO is the aerosol microphysical properties, notably the single scattering albedo (SSA) that may not be well represented in both the retrieval and GEOS-5, which could lead to an artificial inflation of the retrieval accuracy metrics, and potentially leading to inaccuracies in model assimilated fields. For the SEAO, it has been found that the SSA is much more variable, both dependent on the measurement method, and from case study to case study (See Pistone et al., 2019). Differences in MOD06ACAERO and ORACLES data can be partially attributed to the underlying aerosol model (SSA) used in MOD06ACAERO (see LeBlanc et al., 2020). This potential problem is identified in Line 305-306, but elaboration of the impacts could be expanded. Inclusion of a figure showcasing the differences in Single Scattering Albedo output from GEOS-5 and used in MCARS as compared to the model from MOD06ACAERO (MOD04_DT), might be instructive here with an option to include the spread of SSA either from SAFARI2000 or from ORACLES as presented for few cases in Pistone et al., 2019.

*This paper is a continuation of explorations in Wind et al 2013 and 2016. The differences between those specific aerosol models had been examined in great detail in Wind et al, 2016. Therefore we did not feel that a repeat discussion was warranted in this case. Additional reference to Wind et al 2016 has been added to the text in order to clarify.*

General Comments:

1. Of general interests, and idea for future directions: Can MCARS be used to evaluate the emission sources (maybe as represented as MODIS fire counts)?

*Absolutely. If a model that serves as input to the simulation models specific emission sources as to location, amount and type of any particulate and radiative emission and any surface properties of emission source, those sources will be represented in the simulation. Our present simulations do not include fire/burned area information. Such information of course can be included if there is interest and funding for evaluation of an algorithm like MODIS Active Fire.*

Specific Comments:

1. Line 62: There is evidence that neighboring clear sky AOD retrievals match the above cloud AOD: Shinozuka et al., 2020

*Indeed there is. However that information would not be of much help in a situation presented in Figure 2. With average cloud cover in that specific area hovering between 60 and 80% during the burning season, it is difficult to perform clear-sky retrievals as standard clear-sky MODIS Aerosol product requires a 10x10 km box of clear sky pixels with the additional included retrieval requiring a 3x3km box of clear sky pixels. Such conditions often cannot be met when marine stratus deck is present in the area. Additional clarifying text has been added.*

2. Line 71: please add the reference in the parenthetical "(add reference)"

*Corrected. Thank you for pointing out the error.*

3. Line 77-78: From reading of the figure it looks closer to 70-80% and 30-50% for June and September respectively.

*Made corrections accordingly. Thank you very much.*

4. Line 96: Redemann et al., citation can be updated to the overview paper: "Redemann, J., Wood, R., Zuidema, P., Doherty, S., Luna, B., LeBlanc, S., Diamond, M., Shinozuka, Y., Chang, I., Ueyama, R., Pfister, L., Ryoo, J., Dobracki, A., da Silva, A., Longo, K., Kacenelenbogen, M., Flynn, C., Pistone, K., Knox, N., Piketh, S., Haywood, J., Formenti, P., Mallet, M., Stier, P., Ackerman, A., Bauer, S., Fridlind, A., Carmichael, G., Saide, P., Ferrada, G., Howell, S., Freitag, S., Cairns, B., Holben, B., Knobelspiesse, K., Tanelli, S., L'Ecuyer, T., Dzambo, A., Sy, O., McFarquhar, G., Poellot, M., Gupta, S., O'Brien, J., Nenes, A., Kacarab, M., Wong, J., Small-Griswold, J., Thornhill, K., Noone, D., Podolske, J., Schmidt, K. S., Pilewskie, P., Chen, H., Cochrane, S., Sedlacek, A., Lang, T., Stith, E., Segal-Rozenhaimer, M., Ferrare, R., Burton, S., Hostetler, C., Diner, D., Platnick, S., Myers, J., Meyer, K., Spangenberg, D., Maring, H. and Gao, L.: An overview of the ORACLES (ObseRvations of Aerosols above CLouds and their intEractionS) project: aerosol-cloud-radiation interactions in the Southeast Atlantic basin, Atmos. Chem. Phys., 21, 1507–1563, doi:10.5194/acp-21-1507-2021, 2021."

*Done. Thank you very much.*

5. Line 185-186, Why were the resolution of streams in DISORT increased from the somewhat standard 16 to 32or even 64? This change increases computation time, so it is likely needed to better resolve the radiances, but what was the metric used to determine this need. (especially for outside of the cloud bow region)

*Both MOD06 and MOD06ACAERO forward lookup tables are generated at 64 streams. We increased resolution to 64 streams for one specific case where cloud bow effect was most visible in order to see if that artifact was due to insufficient computational resolution of the simulation. That turned out not to be the case. The rest of the simulations had been performed at 32 streams. We chose to generally increase the computational resolution over previous MCARS versions that ran at 16 streams in order to better resolve cloud phase functions as better computational resources became available and we could keep the same wall clock time while computing at higher resolution. MCARS is highly flexible and we can change computational resolution at will.*

6. Line 265: dataset number of points at 13.5 millions seems to present an inflated sense of statistics, it would be interesting to report the number of successful MOD06ACAERO retrievals that are used in the comparison.

*MOD06ACAERO retrievals were attempted over each of the 13.5 million points. They were of course not always successful. It can be an interesting side study to look at the failures of retrievals attempted and discern reasons as to why they failed. In order to do so however, no*

*new simulation data would need to be generated. The text has been updated to clarify that 13.5 million points refers to number of simulated radiances with known properties and does not mean that there were 13 million successful MOD06ACAERO retrievals. Number of points used for statistics in figures 4-10 is indicated on each figure.*

7. Figure 4-10: The 2 different panels seem to have differences in their representative color levels. Inclusion of a colorbar would be useful here.

*All 2D histograms are normalized between 0 and 1. Figures have been updated to include the colorbar. Thank you very much.*

References:

LeBlanc, S. E., Redemann, J., Flynn, C., Pistone, K., Kacenelenbogen, M., Segal-rosenheimer, M., Shinozuka, Y., Dunagan, S., Dahlgren, R. P., Meyer, K., Podolske, J., Howell, S. G., Freitag, S., Small-griswold, J., Holben, B., Diamond, M., Wood, R., Formenti, P., Piketh, S., Maggs-Kölling, G., Gerber, M. and Namwoonde, A.: Above-cloud aerosol optical depth from airborne observations in the southeast Atlantic, Atmos. Chem. Phys., 20, 1565–1590, doi:10.5194/acp-20-1565-2020, 2020.

Pistone, K., Redemann, J., Doherty, S., Zuidema, P., Burton, S., Cairns, B., Cochrane, S., Ferrare, R., Flynn, C., Freitag, S., Howell, S. G., Kacenelenbogen, M., LeBlanc, S., Liu, X., Schmidt, K. S., III, A. J. S., Segal-Rozenhaimer, M., Shinozuka, Y., Stamnes, S., van Diedenhoven, B., Van Harten, G. and Xu, F.: Intercomparison of biomass burning aerosol optical properties from in situ and remote-sensing instruments in ORACLES-2016, Atmos. Chem. Phys., 19, 9181–9208, doi:10.5194/acp-19-9181-2019, 2019.

Shinozuka, Y., Kacenelenbogen, M. S., Burton, S. P., Howell, S. G., Zuidema, P., Ferrare, R. A., LeBlanc, S. E., Pistone, K., Broccardo, S., Redemann, J., Schmidt, K. S., Cochrane, S. P., Fenn, M., Freitag, S., Dobracki, A., Segal-Rosenheimer, M. and Flynn, C. J.: Daytime aerosol optical depth above low-level clouds is similar to that in adjacent clear skies at the same heights: airborne observation above the southeast Atlantic, Atmos. Chem. Phys., 20(19), 11275–11285, doi:10.5194/acp-20-11275-2020, 2020.

*We added the references suggested. Thank you very much.*

---

## Author Comment (AC3)

Review comments on 'Analysis of the MODIS Above-Cloud Aerosol Retrieval Algorithm Using MCARS' by Wind et al.

This manuscript uses Multi-sensor Cloud and Aerosol Retrieval Simulator (MCARS) and GEOS5 Nature Run (G5NR) model output to simulate the MODIS radiance, run above cloud aerosol retrieval using generated radiance, and evaluate the MODIS above cloud aerosol retrieval product (MOD06ACAERO), focusing on the absorbing aerosol above low-level water cloud scene from Southeast Atlantic Ocean. The community need for such an aerosol product is urgent since it fills the aerosol data gap in regular MODIS products, which can only retrieve aerosol in the clear sky and on a larger scale (i.e., 3X3 or 10X10 km). Such a 'truth in truth out' study can validate the instrument performance, improve retrieval algorithm, and investigate the potential uncertainties in the retrieval product. This method is beneficial, especially in future satellite missions. Overall, this is a well-written paper with a clear description of the data and methods, and the results are presented and discussed in an appropriate and concise manner. This paper should be accepted essentially with one major comment and few minor revisions discussed below.

Major comment:

One potential uncertainty in aerosol retrieval comes from the aerosol model, especially the single scattering albedo (SSA). The MOD06ACAERO algorithm uses the standard MODIS dark target aerosol model. However, this SSA is higher than the ORACLES campaign reported values (Pistone et al., 2019). In a recent GRL paper by Chang et al. 2021, they compared the MOD06ACAERO product with their own MODIS above cloud AOD retrieval and many other observations (e.g., HSRL2, 4STAR) from the ORACLES campaign. And they found that the difference in SSA is a major source of uncertainty in above cloud aerosol retrieval. What is the spectral SSA in the dark target aerosol model, and how it compares to G5NR SSA and real SSA measured from the filed campaign? The correlation between MOD06ACAERO and G5NR ACAOD can be high as long as both aerosol models are similar to each other, but this doesn't mean they are accurate or close to the observations. Therefore, the authors need to address this problem in the paper. And for any algorithm development study, a validation by real observations would be helpful.

*Differences in SSA (aerosol model) are of course the primary source of uncertainty/biases in retrieved above-cloud AOD (Meyer et al, 2016). That said, one of the main objectives of this study is to advise model developers on assimilation constraints for MODACAERO retrievals. The second objective is examine MOD06ACAERO performance on synthetic data with known truth, whatever that truth happens to be. Detailed comparison of aerosol models used by MOD04_DT (and MODACAERO) and the ones used by GEOS-5 was performed in Wind et al*

*(2016), which is listed as a companion paper to this one. Therefore authors did not feel it necessary to repeat the study. But, as the reviewer points out, biases/uncertainties may be present when MODACAERO is running on real data. Estimates of the aerosol model uncertainty are included in the pixel-level retrieval uncertainty reported by the MODACAERO algorithm, and we use that uncertainty to screen the pixels in order to inform data assimilation.*

[Figure]

**Figure 9.** OPAC single scattering albedo as a function of humidity (color) and wavelength. The various relative humidity levels are in order (red, orange, green, and blue) for 95, 80, 30, and 0 % column relative humidity.

*Figure from Wind et al, 2016 showing the MCARS single scattering albedo for absorbing aerosols, which is a dynamic function of column relative humidity. MOD04_DT and MODACAERO use the absorbing aerosol model at constant humidity of 80% (orange line in this plot).*

Specific comments:

1. Please add color bars to figures 4-10.

*Done. Thank  you very much.*

2. Line 71: Reference is missing as authors marked in the text.

*Corrected. Thank you very much.*

3. Line 96: The reference Redemann et al. 2019 may not be appropriate. Consider citing the ORACLES overview paper by Redemann et al. 2021.

*Corrected. Thank you very much.*

4. Line 110: In MCARS simulation, is instrument measurement uncertainty also accounted?

*The instrument measurement uncertainty is accounted for as if it were MODIS instrument. It is explicitly accounted for during retrieval and is a component of overall uncertainty by which the retrievals are screened for statistics in figures 4-10. ACAERO uncertainty components are described in detail in Meyer et al (2013, 2015). There is no additional explicit uncertainty source in this particular MCARS simulation. MCARS has a capability of utilizing various real instrument characteristics, including real band uncertainty models. Looking at simulating various types of instrument uncertainty and measurement degradation impact on retrievals is a planned future study.*

5. Line 118: Increase the number of streams will increase the accuracy of simulated radiance. Meanwhile, the computational time also increases, but how much more accuracy gain due to this change?

*It is a fact that increasing the number of streams leads to better resolution of cloud phase functions specifically. Unfortunately even temporarily increasing resolution to 64 streams did not resolve MODACAERO retrieval issue around the cloud bow. Authors of the MODACAERO algorithm plan further study of that particular area of retrieval space. Such investigation is outside the scope of this study. Here we highlight the issue and check a couple of so-called low-hanging fruits, computational resolution being one of them. Due to increasing power of available computational resources, we made the decision to increase the MCARS resolution from 16 streams to 32, without increasing the wall-clock-time. The authors had been previously told that 16 streams is insufficient for good representation of clouds. MOD06 and MODACAERO forward lookup tables are computed at 64 streams, with MOD06 additionally utilizing the phase function directly to compute the single-scattering component of reflectance.*

6. Line 136: It will be great to compare with the field campaign data (e.g., NASA ORACLES).

*Indeed, and this will be a future focus of the authors of MODACAERO algorithm, especially so that now MOD06ACAERO is an official publicly available MODIS and VIIRS product. This particular study was performed in order to inform model developers as for best practices for*

*assimilation of MODACAERO as it presently stands, as there is a significant amount of interest in the modeling community in assimilation of MODACAERO data. This study also serves to inform MODACAERO developers as to presence of any systemic issues in the algorithm, such as a closure issue. With the exception of cloud bow issue, there does not appear to be one. Armed with that information, the authors of MODACAERO would be able to better evaluate the performance of their algorithm when comparing to ORACLES in-situ measurements.*

7.  Line 270: 'from' is used twice, please delete one.

*Corrected. Thank you very much.*

References:

Chang, I., Gao, L., Burton, S. P., Chen, H., Diamond, M. S., Ferrare, R. A., et al. (2021). Spatiotemporal heterogeneity of aerosol and cloud properties over the southeast Atlantic: An observational analysis. Geophysical Research Letters, 48, e2020GL091469. https://doi.org/10.1029/2020GL091469

Redemann, J., Wood, R., Zuidema, P., Doherty, S. J., Luna, B., LeBlanc, S. E., Diamond, M. S., Shinozuka, Y., Chang, I. Y., Ueyama, R., Pfister, L., Ryoo, J.-M., Dobracki, A. N., da Silva, A. M., Longo, K. M., Kacenelenbogen, M. S., Flynn, C. J., Pistone, K., Knox, N. M., Piketh, S. J., Haywood, J. M., Formenti, P., Mallet, M., Stier, P., Ackerman, A. S., Bauer, S. E., Fridlind, A. M., Carmichael, G. R., Saide, P. E., Ferrada, G. A., Howell, S. G., Freitag, S., Cairns, B., Holben, B. N., Knobelspiesse, K. D., Tanelli, S., L'Ecuyer, T. S., Dzambo, A. M., Sy, O. O., McFarquhar, G. M., Poellot, M. R., Gupta, S., O'Brien, J. R., Nenes, A., Kacarab, M., Wong, J. P. S., Small-Griswold, J. D., Thornhill, K. L., Noone, D., Podolske, J. R., Schmidt, K. S., Pilewskie, P., Chen, H., Cochrane, S. P., Sedlacek, A. J., Lang, T. J., Stith, E., Segal-Rozenhaimer, M., Ferrare, R. A., Burton, S. P., Hostetler, C. A., Diner, D. J., Seidel, F. C., Platnick, S. E., Myers, J. S., Meyer, K. G., Spangenberg, D. A., Maring, H., and Gao, L.: An overview of the ORACLES (ObseRvations of Aerosols above CLouds and their intEractionS) project: aerosol–cloud–radiation interactions in the southeast Atlantic basin, Atmos. Chem. Phys., 21, 1507–1563, https://doi.org/10.5194/acp-21-1507-2021, 2021.

Pistone, K., Redemann, J., Doherty, S., Zuidema, P., Burton, S., Cairns, B., Cochrane, S., Ferrare, R., Flynn, C., Freitag, S., Howell, S. G., Kacenelenbogen, M., LeBlanc, S., Liu, X., Schmidt, K. S., Sedlacek III, A. J., Segal-Rozenhaimer, M., Shinozuka, Y., Stamnes, S., van Diedenhoven, B., Van Harten, G., and Xu, F.: Intercomparison of biomass burning aerosol optical properties from in situ and remote-sensing instruments in ORACLES-

2016, Atmos. Chem. Phys., 19, 9181–9208, https://doi.org/10.5194/acp-19-9181-2019, 2019.

*We added the suggested references. Thank you very much.*

---

## Author Comment (AC4)

This manuscript evaluates the performance of MODIS Above-Cloud AEROsol retrieval algorithm (MOD06ACAERO) in the Southeastern Atlantic for biomass burning aerosols using Multi-Sensor Cloud and Aerosol Retrieval simulator (MCARS). Lack of aerosol retrievals in cloudy conditions is a well-known problem in the aerosol community. This study helps address this issue using a closure study of MOD06ACAERO above-cloud Aerosol Optical Depth (AOD). The paper meticulously describes the observations and model and discusses the limitations and strengths of the retrieved AOD for different filtering conditions based on cloud cover, zenith angle, and pixel-level retrieval uncertainty. The results from this study are beneficial for model evaluation studies and design of future satellite missions.

Filtering conditions and related error metrics from this study provide an opportunity to test the model performance from assimilation of MOD06ACAERO AOD retrievals. Availability of code and data will make it easier to test this approach for other satellite instruments and aerosol types. The paper is well-written and easy to read. However, there are some major revisions that need to be addressed before the paper is published.

Major Revisions:

- Differences in the aerosol model between MOD06ACAERO retrievals and GEOS-5 need to be discussed in the analysis section. This can be included before describing the results from the sensitivity tests (before line 290).

*A detailed study of those differences is presented in Wind et al (2016). Text has been amended to clarify.*

- What are the uncertainties/biases in the retrieved AOT (using MCARS) from this study? What are the factors influencing the uncertainties over land and ocean for the retrieved AOT using MCARS synthetic radiances? It is not clear if the algorithm is useful for clear-sky conditions or can be used only in cloudy conditions.

*MO06ACAERO algorithm is to be applied strictly over ocean. It does not execute over land. The algorithm is useful for both clear-sky and cloudy conditions and it was evaluated for both in this study. For detailed information about this algorithm one must consult Meyer et al (2013 and 2015). This paper gives only a brief overview of the algorithm details, as that is all that is needed here. Here the focus is on the closure and performance over synthetic data only.*

- Figure 2. Shows real data example of MOD06ACAERO Aerosol Optical Depth (AOD) retrieval and MODIS Dark-Target (DT) aerosol retrieval. It is clear that

the data gaps are reduced in panel c). However, there are differences in AOD values between panel b) and c) even in areas where MODIS DT clear-sky retrievals are available. What are the reasons for these differences?

*These differences are due primarily to aerosol radiative model differences and the different sensitivities of the retrieval approaches (clear sky retrievals are generally sensitive to aerosol scattering, while above-cloud retrievals are sensitive to absorption). While MOD06ACAERO uses the absorbing aerosol model from the MOD04 Dark Target over land retrievals, MOD04 uses a different retrieval approach over oceans that combines various course and fine mode aerosol models that are different from their land assumptions.*

- How does the retrieved AOD compare against MODIS DT in clear-sky conditions?

*This information can be found in Meyer et al (2013 and 2015), with additional comparisons against CALIPSO. The agreement was very good, considering that MOD06ACAERO and MOD04_DT use the same exact set of aerosol models. This particular study is a closure study where the algorithm is tested on a synthetic dataset with known "truth" for every pixel. Comparisons of MOD06ACAERO results with other sensors are outside the scope of this study.*

- Figure 4-10 – it is not shown what do colors represent in the panels. Do they represent probability density values? Adding a colorbar and description of the colorbar in the caption is necessary.

*Corrected. Thank you very much.*

Minor Revisions:

- In general, it will be interesting to assess the performance of retrieved AOD in the Atlantic for dust transport from Africa above the clouds. For this paper, it will be helpful to include comments/references on the performance of MOD06ACAERO for other aerosol types.

*MOD06ACAERO is a strictly regional algorithm developed for use in SEAO and similar areas around the globe. There are presently no plans to use it anywhere else. Any such studies would be in the domain of authors of MOD06ACAERO algorithm and are outside the scope of this paper. This study focuses on developing good practices for the modeling community for use of data that the authors of MOD06ACAERO might consider for public release.*

- Although the goal of this paper is to evaluate MOD06ACAERO, comparison of the results against ORACLES data will strengthen the paper.

*The goal of this paper is to develop a set of assimilation constraints for model developers interested in assimilating MODACAERO retrievals into their models. Comparisons of MODACAERO retrievals with in-situ field campaign data are planned by the authors of MODACAERO algorithm as a separate and unrelated study. Text has been amended to clarify that point.*

- Line 71 add reference – reference is missing. "... daily mean cloud fraction greater than 50% in the MODIS Daily Level-3 gridded product (add reference)"

*Corrected. Thank you very much.  .*

- Lines 277-280 : It is understandable that G5NR is a free running model and any resemblance to real data is a coincidence. Please elaborate (or rephrase) lines 277-280 to include comments on a similar comparison of cloud amount/distribution in MODIS granules and G5NR. Spatial distribution of clouds and cloud optical properties between G5NR and MODIS granules can affect the retrieved AOD. How is this addressed? (see Le Blanc et al., 2020).

*Cloud amount and distribution are not important in this case as no intent of comparison with any real data is made at any point. That statement is made in text a number of times. What is important, that however clouds and aerosols are distributed and whatever their amount is, the content of atmospheric column for every pixel for which MOD06ACAERO retrieval is attempted is a known quantity at all times. We examine how well the MOD06ACAERO code is able to retrieve that "known quantity".*

- Line 270 – "from from the simulation offers". Remove one.

*Corrected. Thank you very much.*

- Figure 4-10 - colorbar is missing. Since the data density between panels a and b change in these figures, it would be more meaningful to include normalized error metrics in these figures such as normalized mean bias or normalized RMSE, fractional gross error. Please add what colorbar represents in the figure captions (perhaps, probability densiity values) ?

*Colorbars have been added. The color scale is normalized density. RMSE and fit equation are present on every panel of every histogram.*

---

## Referee Report (RR1)

**Round 2 Review of "Analysis of the MODIS Above-Cloud Aerosol Retrieval Algorithm Using MCARS" by Wind et al.**

*Summary:*

The manuscript revisions are very good, particularly for the specific comments. That said the major comment identified by all three reviewers may still need a minor revision to be adequately addressed.

The major comment relies on the aerosol model differences in MCARS and MOD06ACAERO, particularly the Single Scattering Albedo (SSA). It is understood that the author's response was to highlight the work described by Wind et al. (2016).

*Line 316-318:*

"Detailed comparison of GOCART and MOD04_DT aerosol models for biomass burning aerosols has been performed in Wind et al (2016)."

Expansion of this sentence with a summary of the impacts of AOD matching found by Wind et al. (2016) here would be appropriate. Of interest would be the selected case study (fig. 9 in Wind et al. 2016) shows higher SSA values than what would be expected for southeast Atlantic (e.g., Pistone et al., 2019), and any conjecture on possible impact to AOD.

References:

Pistone, K., Redemann, J., Doherty, S., Zuidema, P., Burton, S., Cairns, B., Cochrane, S., Ferrare, R., Flynn, C., Freitag, S., Howell, S. G., Kacenelenbogen, M., LeBlanc, S., Liu, X., Schmidt, K. S., III, A. J. S., Segal-Rozenhaimer, M., Shinozuka, Y., Stamnes, S., van Diedenhoven, B., Van Harten, G. and Xu, F.: Intercomparison of biomass burning aerosol optical properties from in situ and remote-sensing instruments in ORACLES-2016, Atmos. Chem. Phys., 19, 9181–9208, doi:10.5194/acp-19-9181-2019, 2019.

---

## Author Response (AR2)

Text has been altered as follows:

MCARS has the ability to switch between the GEOS-5 aerosols and those used by MOD06ACAERO and MOD04DT. We tested part of the dataset with identical aerosol models between retrieval and simulation and found there to be no significant impact. One reason for that is simulations in Wind et al (2016) dealt with aerosols located near sources. These aerosols, even though they are same basic type, traveled a significant distance from source and have had a chance to absorb water. Once that happens, there is no difference in the scattering properties between the aerosol model used by MOD04DT and GEOS-5. Part of the reason of this specific dataset selection is to also have the cloud-free land present so that we could repeat the experiment in Wind et al (2016) on a different continent. We expect over land, and thus near sources, we would absolutely see the impact of differences in single scattering albedo.